# Malondialdehyde as a Useful Biomarker of Low Hand Grip Strength in Community-Dwelling Stroke Patients

**DOI:** 10.3390/ijerph17217918

**Published:** 2020-10-28

**Authors:** Onchuma Mueangson, Parinya Vongvaivanichakul, Kornyok Kamdee, Chutima Jansakun, Wanatsanan Chulrik, Pongphan Pongpanitanont, Pornchai Sathirapanya, Warangkana Chunglok

**Affiliations:** 1School of Allied Health Sciences and Research Institute for Health Sciences, Walailak University, Nakhon Si Thammarat 80161, Thailand; onchuma.mu@wu.ac.th (O.M.); parinya.vo@wu.ac.th (P.V.); kornyok.ka@wu.ac.th (K.K.); chutimaj65@gmail.com (C.J.); manatwanatsananmt@gmail.com (W.C.); pongphan.po@wu.ac.th (P.P.); 2Division of Neurology, Department of Internal Medicine, Faculty of Medicine, Prince of Songkla University, Songkla 90110, Thailand; sporncha@medicine.psu.ac.th

**Keywords:** ischemic stroke, muscle strength, hydrogen peroxide, oxidative damage, inflammatory cytokines

## Abstract

The assessment of muscle strength by hand grip strength (HGS) is used to evaluate muscle weakness and wasting among stroke patients. This study aimed to investigate the association of oxidative stress/oxidative damage and inflammatory biomarkers with muscle strength and wasting, as evaluated by HGS, among community-dwelling post-stroke patients. The HGS of both paretic and non-paretic limbs was negatively associated with modified Rankin scale (mRS) values. The serum levels of catalase activity and malondialdehyde (MDA), and plasma tumor necrosis factor (TNF)-α levels were significantly increased in post-stroke patients compared with non-stroke controls. Further analysis highlighted that hydrogen peroxide was positively correlated with HGS in the paretic limbs. Interestingly, an elevated MDA level, excluding advanced age and high mRS, increased the risk of low HGS in the non-paretic limbs of stroke patients. This study suggests that there is a detrimental association between MDA and muscle strength and early muscle wasting among post-stroke patients. Hence, MDA is a potentially useful biomarker of muscle weakness and wasting in post-stroke patients living in the community.

## 1. Introduction

Strokes place a major burden on the health care system globally [1]. The most frequent impairment after stroke is hemiparesis, followed by dysarthria, visual disturbance, facial palsy, spasticity, imbalance, post-stroke pain or physical discomfort, and dysphagia. These conditions significantly affect stroke patients’ activities of daily living as well as their quality of life [2,3]. Changes in the size and strength of the skeletal muscles in both paretic and non-paretic limbs have been reported in post-stroke patients [4]. As early as 4 to 30 h after the onset of an acute ischemic stroke (IS), the number of motor units estimated in the hypothenar muscles of paretic limbs starts to decline compared with that in non-paretic limbs [5]. An ongoing reduction in the number of motor units, a disturbance in the motor unit firing intensity, and long-standing muscle denervation were observed in the paretic muscles in late-stage post-stroke subjects. These factors could contribute to muscle weakness and wasting [6,7]. Notably, muscle weakness has also been shown to appear in the non-paretic limbs within the first week after the onset of a stroke [8].

Hand grip strength (HGS) is widely accepted as a reliable measurement for whole-body muscle strength in healthy adults, as well as an indicator of muscle weakness and wasting after stroke [9,10,11]. The relative sustainability of the paretic side’s grip force was shown to be 20–30% lower than that of the non-paretic one at 2–4 weeks post-stroke, and the value gradually rose to a level comparable with that of the non-paretic side at 6 months post-stroke [12]. The HGS of the non-paretic hands in stroke patients was found to be obviously less than the HGS of the dominant hand in non-stroke controls [13]. Furthermore, the HGS on the non-paretic side has been shown to be positively associated with the competence of pulmonary function and respiratory muscle strength [14], functional improvement after rehabilitation [15], and the degree of independency, as evaluated by the Barthel index (BI) at hospital discharge [16] in late post-stroke patients. In contrast, the HGS of the paretic hand was negatively correlated with the stroke severity, as evaluated by the National Institutes of Health Stroke Scale (NIHSS) score, and the degree of disability, as assessed by the modified Rankin Scale (mRS) at discharge and 3 months after stroke occurrence [16]. Recently, the Asian Working Group for Sarcopenia (AWGS) 2019 suggested that a low HGS is one of the key measurements of “possible sarcopenia” used in primary or community health care services [17]. The Asian Working Group for Sarcopenia (AWGS) 2014 defined “sarcopenia” as age-related loss of muscle mass together with decreased muscle strength and/or loss of physical performance [18]. The term was used to describe the loss of muscle mass and functions related to various post-stroke illnesses and muscle disuse conditions [19,20].

Oxidative stress/oxidative damage and inflammation play roles in the pathogenesis of stroke-induced muscle weakness and eventual wasting and disability [21]. Most published articles reveal a correlation of oxidative stress/oxidative damage and inflammatory biomarkers, e.g., malondialdehyde (MDA), 4-hydroxy-2 (E)-nonenal (4-HNE), the erythrocyte sedimentation rate (ESR), C-reactive protein (CRP), and the inflammatory cytokines interleukin (IL)-6 and tumor necrosis factor (TNF)-α with sarcopenia in older adults [22,23,24]. To date, only a few studies have investigated the correlations of the aforementioned biomarkers with muscle wasting in post-stroke patients [25]. A study reported that the presence of systemic inflammation determined by CRP is negatively associated with HGS and BI but positively associated with mRS [26]. Furthermore, another study reported that paretic muscles express significantly increased TNF-a mRNA levels compared with the non-paretic muscles of late IS patients, suggesting the presence of active inflammation in paretic muscles [27]. Therefore, this study aims to identify reliable biomarkers that correlate well with low HGS in community post-stroke patients. We propose that low HGS could be associated with higher levels of oxidative stress/oxidative damage biomarkers and inflammatory cytokines.

## 2. Materials and Methods

### 2.1. Study Participants

A total of 338 participants, comprised of 146 IS patients and 192 non-stroke controls, were enrolled in this community-based, cross-sectional study in January 2018 and December 2018. All participants were Buddhists and Muslims who lived in Thasala and Mueang Districts, Nakhon Si Thammarat, Thailand. We enrolled all IS patients who were (1) aged 45–95 years; (2) hemiparetic from first-ever stroke, as diagnosed by a computerized tomography (CT) scan and/or magnetic resonance imaging (MRI) within 24 h of symptom onset; (3) had ≥ 3 months of a stable medical condition post-stroke and received the same class of drug regimen; and (4) had spastic hypertonia in the hand muscles on the paretic side, as determined by a modified Ashworth Scale (MAS) score of < 3. Those with systemic inflammatory diseases, autoimmune diseases, renal or liver diseases, myocardial infarction, cancer, or thyroid diseases; those who were post-surgery within 60 days of enrollment; those unable to communicate well enough to complete the study process; and those that did not provide informed consent were excluded from the study. Non-stroke controls were individuals with no known serious underlying diseases or physical disabilities who were selected from attendants to routine physical check-ups in the sub-district health promoting hospitals in the same areas as the stroke patients.

We used a matched study design (1:1) considering age (±4 years), gender, religion, and body mass index (BMI) (±2.5 units) to evaluate the significant differences between cases and controls. We enrolled the stroke patients and matched controls from non-stroke study participants consecutively.

Data collected included baseline characteristics of the study population from personal interviews by well-trained interviewers using specific questionnaires. The underlying cardiovascular risks based on primary care physician diagnoses in sub-district health promoting hospitals were identified. 

The terms and medical conditions used in the current study were defined. BMI was calculated from weight in kilograms divided by height in meters squared (kg/m^2^) [28]. Adequate physical activity was defined based on participants’ self-reports as exercise ≥ 3 times per week for at least 30 min at a time. Hypertension was defined as a resting systolic blood pressure (SBP) ≥ 140 mmHg and/or diastolic blood pressure (DBP) ≥ 90 mmHg or requirement for antihypertensive drugs [29]. Hyperlipidemia was diagnosed in individuals taking any antilipidemic drugs or acquiring one or more of the following blood lipid results: total cholesterol (TC) ≥ 200 mg/dL, low-density lipoprotein cholesterol (LDL-C) ≥ 130 mg/dL, high-density lipoprotein cholesterol (HDL-C) ≤ 40 mg/dL, or triglycerides (TG) ≥ 150 mg/dL [30]. Diabetes mellitus was taken as a fasting plasma glucose (FPG) ≥ 126 mg/dL or current use of glycemic control medication. Impaired fasting glucose was defined as fasting blood sugar (FBS) of 100–125 mg/dL [31]. Smoking was defined as smoking an average of ≥ 1 cigarette per day in the last 3 months. Alcohol beverage drinking was based on participants’ self-reports of drinking ≥ 100 mL of alcohol more than three times per week. 

The research protocol was approved by the Walailak University ethics committee (16/037), and written informed consent was obtained from all participants. 

### 2.2. Physical Measurement Tools 

#### 2.2.1. Barthel Index (BI)

Stroke patients were evaluated on their ability to perform fifteen tasks independently. These tasks included three main functions: self-care, continence of bowel and bladder, and mobility [32]. The scores were classified as 0–20, 21–60, 61–90, and 91–99, indicating total, severe, moderate, and slight dependency, respectively [33]. 

#### 2.2.2. Modified Rankin Scale (mRS)

The stroke patients were interviewed about their ability to perform their daily activities. The mRS was used to evaluate the degree of disability in doing daily activities in this study. Scores of 0–2 indicated no symptoms to a slight disability, while scores of 3–5 indicated moderate to severe disability, and a score of 6 indicated that the patient was dead [34]. 

#### 2.2.3. Hand Grip Strength (HGS)

HGS (kilograms) was measured using a dynamometer (Takei TKK 5001, Takei Scientific Instruments Co. Ltd., Tokyo, Japan) [35]. While in standing or sitting upright positions, participants had each arm in flexion and did not contact the trunk. Then, they were asked to perform maximal voluntary hand muscle contractions with the dynamometer twice, 2 min apart to avoid muscle fatigue, with each hand. Non-stroke controls were asked to perform the maximal force contractions with the dominant hand. The term “HGS” was the mean values of hand grip strength, and the term “low HGS” was defined as HGS of lower than 26 kg for men and 18 kg for women [18]. 

### 2.3. Blood Sample Processing

Ten milliliters of fasting venous blood was only once at admission collected from each participant and then centrifuged at 1200× *g* for 5 min at 25 °C. Serum and plasma were separated and stored at −80 °C. The obtained serum was assessed for levels of fasting blood sugar (FBS), lipid profile, and oxidative stress/oxidative damage markers (catalase (CAT), superoxide dismutase (SOD), hydrogen peroxide, and MDA). Additionally, plasma from ethylenediaminetetraacetic acid (EDTA) blood samples was assessed for inflammatory cytokines (TNF-α and IL-6). All biomarkers and cytokines were analyzed within 2 weeks after collection.

### 2.4. Measurement of Biomarkers 

#### 2.4.1. Metabolic Profiles

FBS, TG, TC, LDL-c, and HDL-c were measured by an automated analyzer (Beckman Coulter, Brea, CA, USA).

#### 2.4.2. Oxidative Stress/Oxidative Damage 

SOD and CAT activities in the collected serum were determined by a commercial kit (Sigma-Aldrich, St. Louis, MO, USA), and the hydrogen peroxide concentration was determined by the commercial Pierce Quantitative Peroxide Assay kit (Pierce Biotechnology, Rockford, IL, USA), following the manufacturer’s instructions. The MDA level was determined by the concentration of thiobarbituric acid reactive substance (TBARS) in serum based on MDA and the thiobarbituric acid reaction, as previously described [36]. Briefly, serum was added to sodium dodecyl sulfate and thiobarbituric acid under acidic conditions, and then the reaction mixture was heated at 95 °C for 60 min. After cooling with tap water, butanol:pyridine (15:1) solution was added to the mixture, vortexed, and centrifuged at 2000× *g* for 10 min. The formation of TBARS was measured by using a microplate reader (BioTek Instruments Inc., Winooski, VT, USA) at 532 nm. 

#### 2.4.3. Inflammatory Cytokines

IL-6 and TNF-α levels were measured using ELISA kits (BioLegend, San Diego, CA, USA) in accordance with the manufacturer’s instructions. Plasma was prepared as undiluted samples. Serial dilutions of recombinant IL-6 and TNF-α standards were included on each plate to determine plasma IL-6 and TNF-α concentrations in the samples. Then, plasma IL-6 and TNF-α concentrations in samples were measured by a microplate reader at 450 nm, and the values are reported in pg/mL. 

### 2.5. Statistical Analyses

The software package used to analyze the collected data was SPSS (PASW Statistic, version 16, Chicago, IL, USA). The differences in variables between stroke patients and non-stroke controls were evaluated by chi-square tests, independent *t*-tests, and the Mann–Whitney U test. The presence of a normal distribution was assessed using the Kolmogorov–Smirnov test. Categorical variables are presented as proportions, while continuous variables are reported as either means ± standard deviation (SD) or medians (interquartile range (IQR). To identify significant associations between clinical variables and HGS in stroke participants, multiple linear regression and multiple logistic regression analyses were performed, and *p* < 0.05 indicated significance.

## 3. Results

### 3.1. Study Population 

Twenty-five cases in the patient group were excluded due to a lack of matched controls, while another thirty cases were subsequently excluded because of an inadequate volume of serum obtained for biomarker assessments. Therefore, 91 patients were deemed fit for statistical analysis. 

Stroke patients had higher incidences of hypertension (*p* = 0.005), hyperlipidemia (*p* = 0.002), and diabetes mellitus (*p* = 0.001) than controls. The mean (± SD) BI value in stroke patients was 85.33 (±25.73), and the majority of them (71.43%) showed slight dependency and disability (mRs 0–2). TC and LDL-c concentrations were significantly lower in stroke patients compared with controls (both *p* <0.0001). This may be an effect of the cholesterol-lowering drug used in stroke patients. Interestingly, the levels of MDA and TNF-α were significantly higher in stroke patients (*p* = 0.022 and 0.004, respectively) (Table 1).

### 3.2. Hand Grip Strength of Paretic and Non-Paretic Limbs in Stroke Patients

The overall mean HGS of paretic limbs in stroke patients was 13.49 ± 11.04 kg. However, the mean HGS values of males and females were 16.62 ± 11.53 and 7.71 ± 7.19 kg, respectively—lower than the cut-off values for both genders (<26 kg (male), and <18 kg (female).

The mean HGS values of non-paretic limbs and dominant hands of the two study groups did not differ. Male stroke patients had significantly lower HGS of non-paretic limbs than dominant hands of non-stroke controls (*p* = 0.048). However, the number of stroke patients with low HGS was 57 (62.64%), which was not different from the number of controls (50 subjects, 54.95%) (Table 2).

### 3.3. The Impact of Hydrogen Peroxide on HGS in Paretic Limbs of Stroke Patients

The multiple linear regression analysis was used to analyze age, gender, BMI, mRS, CAT, SOD, hydrogen peroxide, MDA, TNF-α, and IL-6 for factor independently correlated with HGS in non-paretic and paretic limbs of stroke patients. Our results showed that age (beta = −0.346; *p* < 0.0001), female (beta = −0.392; *p* < 0.0001), and mRS (beta = −0.404; *p* < 0.0001) were negatively correlated with HGS of non-paretic limbs. For paretic limbs, HGS was negatively correlated with female (beta = −0.202; *p* < 0.014) and mRS (beta = −0.627; *p* < 0.0001). Nonetheless, HGS of paretic limbs was found positively correlated with the level of hydrogen peroxide (beta = 0.195; *p* = 0.026) in stroke patients (Table 3).

### 3.4. The Impact of Serum MDA on the Risk of Low HGS in Non-Paretic Limbs of Stroke Patients

Multiple logistic regression analysis was performed to identify for factor-independent associations of age, gender, BMI, mRS, CAT, SOD, hydrogen peroxide, MDA, TNF-α, and IL-6 with low HGS in non-paretic limbs. The multivariate analysis showed that older age (odds ratio (OR): 1.131; 95% CI: 1.053 to 1.097), higher mRS (OR: 3.357; 95% CI: 1.678 to 6.714), and MDA (OR: 1.280; 95% CI: 1.024 to 1.600) significantly increased the risk of low HGS in the non-paretic limbs of stroke patients (Table 4).

## 4. Discussion

The present study notably reported, for the first time, a positive association between hydrogen peroxide and HGS in paretic limbs. Moreover, age, mRS, and MDA levels were shown to be the risks of low HGS in non-paretic limbs of post-stroke patients.

We found no difference in the HSG of non-paretic limbs between stroke patients and controls. This is possibly due to the comparable degree of dependency and disability, as assessed by BI and mRS, respectively. A reduction of muscle strength, indicated by low HGS, could be caused by a loss of mitochondrial bioenergetics of the muscles during aging and disuse atrophy after stroke. This proposed pathophysiology was confirmed by a muscle pathology study in healthy older adults with absolute bedridden status [37,38]. High mRS was found to be a risk factor for low HGS in non-paretic limbs in post-stroke patients, which supports this finding. It was also suggested that lower neuronal metabolism in the diseased primary motor cortex possibly contributes to low HGS [39].

MDA is a lipid peroxide derived from the peroxidation of polyunsaturated fatty acids [40] that is significantly increased in the acute phase and persistently sustained in the late phase of stroke [41]. One study showed that the plasma lipid peroxide level was significantly increased over time when compared across subjects in acute, subacute, and chronic phases after stroke [42]. Similar to some studies, our study revealed significantly elevated serum MDA levels in post-stroke patients compared with controls [43,44]. Moreover, an elevated serum MDA level was found to be a significant risk factor for low HGS in the non-paretic limbs of stroke patients. A previous study revealed a negatively correlation between MDA and the musculoskeletal index, and MDA has been accepted as a potential early biomarker of sarcopenia in older adults [22]. Therefore, MDA could be used as a blood biomarker for early diagnosis of muscle strength and muscle wasting loss in post-stroke patients. The predominant source of MDA production in post-stroke patients remains unclear. However, increased lipid peroxide and MDA levels might be due to spontaneous reactive oxygen species (ROS) generation by peripheral phagocytes [42], extracellular production of hydrogen peroxide by paretic muscles during denervation [45], or associated comorbidities, e.g., diabetes, hypertension, and hyperlipidemia [46].

Oxidative stress/oxidative damage has a profound effect on stroke pathogenesis and recurrence [47] and has been used as a marker of improvement after exercise training in post-stroke patients [48]. Based on normal physiology, SOD plays a role in regulating ROS signaling functions with subsequent biological responses [45]. In contrast, during oxidative stress, SOD has a regulatory role whereby it scavenges superoxide anions by generating hydrogen peroxide, which is further converted into hydrogen peroxide by CAT and glutathione peroxidase [49]. In our study, stroke patients had significantly decreased SOD activity levels with a concomitant increase in CAT activity compared with that of controls. Exhaustion may decrease serum SOD activity following prolonged ROS generation during ischemic events [50] and skeletal muscle pathology [51]. A previous study suggested that increased plasma SOD activity occurs after dynamic resistance training in post-stroke patients [48]. Another study revealed that the concentration of CAT in hemolysates was significantly increased in stroke patients at a 3-month follow-up check [52]. Therefore, the high levels of CAT activity observed in our stroke population may have potentiated the antioxidant defense mechanism against the increased production of ROS. This finding is concomitant with our study results showing that hydrogen peroxide tended to be decreased in stroke patients compared with controls. Interestingly, the regression analysis showed that hydrogen peroxide was positively associated with HGS in the paretic limbs of stroke patients. This may be due to the requirement of physiological levels of hydrogen peroxide for maintaining normal contractile activity in skeletal muscles [45]. In a previous study, it was suggested that hydrogen peroxide levels are necessary to maintain static force production or strength in the paretic limbs of stroke patients. In addition, a short period of hydrogen peroxide exposure was found to increase sub-maximal muscle force generation [53]. Nevertheless, a previous study revealed that hydrogen peroxide exposure decreased muscle force by suppressing myosin cross-bridge formation [54].

TNF-α has been proposed as a key biomarker of muscle weakness because of the induction of oxidant production in the muscles, causing myofilament dysfunction [55]. Although TNF-α levels were found to be significantly increased in stroke patients compared with controls, a correlation with low HGS was not found in the current study. Oxidative stress/oxidative damage and inflammation are ongoing processes in the chronic stroke phase. Prolonged oxidative stress/oxidative damage and inflammatory reactions after stroke contribute to cachexia after stroke, as determined by body weight loss, declines in fat and lean mass, low HGS, and eventual low physical performance [26]. The increment of oxidative damage biomarkers like MDA may be associated with reduced muscle strength, possibly leading to muscle mass loss among post-stroke patients. Thus, we suggest that antioxidant treatment and increased physical activity may be useful for preventing muscle wasting in post-stroke patients

The association of oxidative stress/oxidative damage/inflammatory biomarkers with low HGS in community post-stroke patients is first documented in the current study. However, the following limitations should be considered: (1) the small sample size and single-center study nature; (2) the missing data and, in particular, insufficient volume of blood samples for biomarker assessment, may have affected the interpretation of the reported findings; and (3) the specific cut-off values used to indicate low HGS in non-paretic limbs for post-stroke patients, which have not been systemically determined. We suggest that the future research should document the use of appropriate sources for oxidative stress/oxidative damage/inflammatory biomarker assessment, i.e., skeletal muscles, of paretic and/or non-paretic limbs, among stroke patients. Furthermore, the association of biomarkers of muscle wasting or sarcopenia after stroke should be scientifically addressed.

## 5. Conclusions

Our study reported that advanced age, high mRS, and high serum MDA levels are risk factors for low HGS in the non-paretic limbs of post-stroke patients. Due to the association of low HGS in the non-paretic limbs of post-stroke patients with subsequent muscle strength loss and muscle wasting, we suggest that the presence of the mentioned clinical characteristics and high MDA levels are potential predictors muscle weakness and wasting in post-stroke patients. Thus, treatments involving reduction of the oxidation product MDA and encouragement of daily physical activity may delay further progression of muscle wasting in post-stroke patients.

## Figures and Tables

**Table 1 ijerph-17-07918-t001:** Baseline characteristics of non-stroke controls and stroke patients.

Variables	Stroke Patients	Non-Stroke Controls	*p*-Value
	(*n* = 91)	(*n* = 91)	
Age (years)	69 (14)	68 (13)	0.916
Gender, *n* (%)			1
Male	59 (64.84)	59 (64.84)	
Female	32 (35.16)	32 (35.16)	
Body mass index, kg/m^2^	22.51 ± 3.12	22.62 ± 3.53	0.826
Physical activity, *n* (%)			0.432
No	28 (30.77)	33 (36.26)	
Yes	63 (69.23)	58 (63.74)	
Vascular risk factors, *n* (%)			
Hypertension	74 (81.32)	57 (62.64)	0.005 ^b^
Hyperlipidemia	73 (80.22)	54 (59.34)	0.002 ^b^
Diabetes mellitus	26 (28.57)	8 (8.79)	0.001 ^b^
Impaired fasting glucose	9 (9.89)	16 (17.58)	0.132
Smoking status, *n* (%)			0.581
No	71 (78.02)	74 (81.32)	
Yes	20 (21.98)	17 (18.68)	
Alcohol drinking, *n* (%)			0.155
No	84 (92.31)	78 (85.71)	
Yes	7 (7.69)	13 (14.29)	
Barthel index, mean ± SD	85.33 ± 25.73	100.00 ± 0.00	<0.0001 ^a^
0–20, *n* (%)	5 (5.49)	0 (0.00)	
21–60, *n* (%)	13 (14.28)	0 (0.00)	
61–90, *n* (%)	10 (10.99)	0 (0.00)	
91–99, *n* (%)	9 (9.89)	0 (0.00)	
100, (%)	54 (59.34)	91 (100.00)	
Modified Rankin Scale, mean ± SD	1.73 ± 1.28	N/A	N/A
(0–2), *n* (%)	65 (71.43)	N/A	N/A
(3–5), *n* (%)	26 (28.57)	N/A	N/A
Biochemical parameters			
FBS (mg/dL)	95.00 (18.25)	93.00 (14.25)	0.224
TG (mg/dL)	119.00 (89.25)	130.00 (67.50)	0.544
TC (mg/dL)	164.28 ± 41.50	203.20 ± 37.93	<0.0001 ^c^
LDL-c (mg/dL)	98.50 (41.25)	128.80 (48.10)	<0.0001 ^a^
HDL-c (mg/dL)	50.16 ± 12.74	49.67 ± 12.18	0.663
CAT (U/L)	132.21 (98.61)	76.85 (96.80)	<0.0001 ^a^
SOD (U/L)	3149.00 (2350.38)	3791.42 (2716.27)	<0.0001 ^a^
Hydrogen peroxide (µM)	14.40 (7.75)	18.76 (8.88)	0.064
MDA (µM)	7.05 (3.49)	5.98 (5.68)	0.022 ^a^
TNF-α (pg/mL)	37.93 (58.63)	22.17 (21.69)	0.004 ^a^
IL-6 (pg/mL)	27.30 (71.94)	42.51 (88.36)	0.144

CAT, catalase; FBS, fasting blood sugar; HDL-c, high-density lipoprotein cholesterol; IL-6, interleukin 6; LDL-c, low-density lipoprotein cholesterol; MDA, malondialdehyde; N/A, not available; SOD, superoxide dismutase; TC, total cholesterol; TG, triglyceride; TNF-α, tumor necrosis factor-alpha. ^a^
*p* < 0.05, Mann–Whitney U test, ^b^
*p* < 0.05, chi-square test, ^c^
*p* < 0.05, independent *t*-test.

**Table 2 ijerph-17-07918-t002:** Hand grip strength of paretic and non-paretic limbs in stroke patients and dominant hands in non-stroke controls.

Clinical Variables	Stroke Patients		Non-Stroke Controls		*p*-Value
	*n* (%)	Mean ± SD	*n* (%)	Mean ± SD	
Hand grip strength, kg					
(paretic-limbs)					
Total	91 (100.00)	13.49 ± 11.04	N/A	N/A	N/A
Male	59 (67.53)	16.62 ± 11.53	N/A	N/A	N/A
Female	32 (32.47)	7.71 ± 7.19	N/A	N/A	N/A
Hand grip strength, kg					
(non-paretic limbs/dominant hands)					
Total	91 (100.00)	21.24 ± 8.61	91 (100.00)	23.05 ± 8.15	0.148
Male	59 (67.53)	24.62 ± 7.92	59 (64.83)	26.75 ± 7.05	0.127
Female	32 (32.47)	15.01 ± 5.87	32 (35.17)	16.23 ± 5.08	0.380
Low hand grip strength, kg					
(non-paretic limbs/dominant hands)					
Total	57 (62.64)	16.47 ± 5.58	50 (54.95)	17.94 ± 5.14	0.212
Male	35 (38.46)	19.22 ± 4.50	29 (31.87)	21.29 ± 3.33	0.048 *
Female	22 (24.18)	11.99 ± 4.11	21 (23.08)	13.31 ± 3.24	0.895

N/A, not available. * *p* < 0.05, independent *t*-test.

**Table 3 ijerph-17-07918-t003:** Multiple linear regression analysis of hand grip strength with variables in stroke patients.

	Hand Grip Strength					
Variables	Non-Paretic Arm			Paretic Arm		
	B	Beta	*p*-Value	B	Beta	*p-*Value
Age	−0.294	−0.346	<0.0001 *	−0.132	−0.121	0.151
Female ^a^	−7.046	−0.392	<0.0001 *	−4.643	−0.202	0.014
BMI	−0.043	−0.016	0.846	0.108	0.031	0.716
mRS	−2.707	−0.404	<0.0001 *	−5.382	−0.627	<0.0001 *
CAT (U/L)	−0.002	−0.021	0.784	−0.012	−0.095	0.241
SOD (U/L)	0.000	−0.115	0.191	0.000	−0.116	0.207
Hydrogen peroxide (µM)	0.121	0.092	0.272	0.329	0.195	0.026
MDA (µM)	−0.251	−0.101	0.202	0.042	0.013	0.872
TNF-α (pg/mL)	0.009	0.114	0.189	0.005	0.052	0.565
IL-6 (pg/mL)	−0.001	−0.014	0.869	0.000	0.003	0.972

^a^ Males and females were assigned values of 0 and 1, respectively. The *p*-values determined by the linear regression were adjusted for age, gender, body mass index (BMI), modified Rankin Scale (mRS), catalase (CAT), superoxide dismutase (SOD), hydrogen peroxide, malondialdehyde (MDA), tumor necrosis factor (TNF)-α, and interleukin (IL)-6. * *p* < 0.05.

**Table 4 ijerph-17-07918-t004:** Multivariate logistic regression analysis determining variables associated with low hand grip strength risk in stroke patients.

Variables	B	S.E.	Adjusted Odds Ratio	95% Confidence Interval	Adjusted *p*-Value
Age	0.123	0.036	1.131	1.053, 1.097	0.001 *
Female ^a^	0.786	0.673	2.194	0.586, 8.211	0.243
BMI	0.043	0.098	1.044	0.861, 1.266	0.663
mRS	1.211	0.354	3.357	1.678, 6.714	0.001 *
CAT (U/L)	0.003	0.004	1.003	0.996, 1.010	0.427
SOD (U/L)	0.000	0.000	1.000	1.000, 1.001	0.112
Hydrogen peroxide (µM)	−0.073	0.048	0.929	0.846, 1.021	0.127
MDA (µM)	0.274	0.119	1.280	1.024, 1.600	0.030 *
TNF-α (pg/mL)	0.001	0.002	1.001	0.997, 1.004	0.802
IL-6 (pg/mL)	0.000	0.002	1000	0.996, 1.003	0.825

^a^ Males and females were assigned values of 0 and 1, respectively. The *p*-values by logistic regression are adjusted for age, gender, body mass index (BMI), modified Rankin Scale (mRS), catalase (CAT), superoxide dismutase (SOD), hydrogen peroxide, malondialdehyde (MDA), tumor necrosis factor (TNF)-α, and interleukin (IL)-6. * *p* < 0.05.

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
