# Peer review of "Malondialdehyde as a Useful Biomarker of Low Hand Grip Strength in Community-Dwelling Stroke Patients"

_ijerph, 2020, doi:10.3390/ijerph17217918_

Round 1

Reviewer 1 Report

In this manuscript, Mueangson and colleagues investigate the association between oxidative stress and inflammatory markers and muscle strength and wasting in post-stroke patients.

The main finding is that elevated malondialdehyde serum levels positively correlated with hand grip strength in the paretic limbs of patients.

The clinical study is interesting and appears to be correctly performed. However, the authors should take care of the following points:

1. The experiments regarding the hand grip strength tests are quite confusing. Specifically:

a) Because the values are used in Table 3, the mean HGS of paretic limbs in stroke patients should be included in Table 2,

b) The difference between hand grip strength (top of Table) and “low” hand grip strength (bottom) is not explained in the Methods nor the Results section and therefore is unclear,

c) Throughout the manuscript, the terms HGS and low HGS appear to be used randomly, with no clear rationale.

2. The authors should state in the Abstract in which tissue catalase, MDA and TNF-α levels were measured.

Author Response

Please find the attached file of response to reviewer-1

Reviewer 2 Report

This study present data demonstrating that high malondialdehyde (MDA) levels in the blood was one of the independent risk factors of low HGS in non-paretic limbs of post-stroke patients. And it also can be used as a good indicator of the prognosis of patients who underwent ischemic stroke (IS).

Although the manuscript provides striking data that suggests MDA can be a new useful biomarker of muscle weakness post-IS, the results of previous studies have proven inconclusive with several concerns regarding the statistical data and design outlined below:

1.In cross-sectional studies, to estimate the prevalence of unknown parameters from the target population using a random sample, an adequate sample size, calculated by formula based on incidence rate, is needed to estimate the population prevalence with good precision. Then, how did the authors estimate sample sizes in this study? Please describe it.

2.Did the all cytokines and biomarkers were analyzed for the first time of admission?   When the blood samples were collected? Please describe the “Methods” in detail.

3.During the 3 months post-IS, did all patients, enrolled in this study, had the same medication treatment? If not, each type of medication can also be a new parameter that can affect the outcome

Author Response

Please find the attached file of response to reviewer-2
